# Interpretation of morphogen gradients by a synthetic bistable circuit

Paul K. Grant [1✉], Gregory Szep [1,2,9], Om Patange [3,7,8,9], Jacob Halatek [1], Valerie Coppard[1], Attila Csikász-Nagy [2,4], Jim Haseloff [5], James C. W. Locke[1,3,6], Neil Dalchau [1] & Andrew Phillips [1✉]

During development, cells gain positional information through the interpretation of dynamic morphogen gradients. A proposed mechanism for interpreting opposing morphogen gradients is mutual inhibition of downstream transcription factors, but isolating the role of this specific motif within a natural network remains a challenge. Here, we engineer a synthetic morphogen-induced mutual inhibition circuit in *E. coli* populations and show that mutual inhibition alone is sufficient to produce stable domains of gene expression in response to dynamic morphogen gradients, provided the spatial average of the morphogens falls within the region of bistability at the single cell level. When we add sender devices, the resulting patterning circuit produces theoretically predicted self-organised gene expression domains in response to a single gradient. We develop computational models of our synthetic circuits parameterised to timecourse fluorescence data, providing both a theoretical and experimental framework for engineering morphogen-induced spatial patterning in cell populations.

[1] Microsoft Research, 21 Station Road, Cambridge CB1 2FB, UK. [2] Randall Centre for Cell and Molecular Biophysics, King's College London, London WC2R 2LS, UK. [3] Sainsbury Laboratory, University of Cambridge, Cambridge CB2 1LR, UK. [4] Faculty of Information Technology and Bionics, Pázmány Péter Catholic University, Budapest 1083, Hungary. [5] Department of Plant Sciences, University of Cambridge, Cambridge CB2 3EA, UK. [6] Department of Biochemistry, University of Cambridge, Cambridge CB2 1QW, UK. [7] Present address: Department of Molecular Biology, Massachusetts General Hospital, Boston, MA 02114, USA. [8] Present address: Department of Genetics, Harvard Medical School, Boston, MA 02115, USA. [9] These authors contributed equally: Gregory Szep, Om Patange. ✉email: grant.paul@microsoft.com; andrew.phillips@microsoft.com

The positional information solution to the French flag problem, in which cells compute their spatial position by responding to the concentration of a morphogen in a gradient[1], provides crucial insight into how patterns of gene expression form in a developing organism. The simplest formulation of this model – concentration thresholds leading directly to gene expression states – requires a static morphogen gradient to produce a stable pattern of gene expression[2,3]. However, quantitative measurements in developing embryos reveal that morphogen gradients are both dynamic and transient[4,5], and genetic perturbations reveal that pattern formation is robust to changes in morphogen concentration[6–8]. A gene regulatory network topology of mutual inhibition downstream of antiparallel morphogen gradients[9–12] (Fig. 1a) has been proposed to robustly interpret dynamic gradients (Fig. 1b). However, while certain features of this topology are common to a number of developmental contexts such as the early *Drosophila* embryo and the vertebrate neural tube (reviewed in [3]), demonstrating how this network functions and whether it is indeed sufficient remains a challenge, due to the complexities of the different biological contexts in which it operates. Recent work in synthetic biology has proven the utility of building multicellular patterning circuits both for understanding development and for learning engineering principles[13–18].

Here we show that the mutual inhibition motif[19] is sufficient to produce stable domains of gene expression in response to dynamic and transient morphogen gradients. By taking a synthetic biology approach[20–23] we have built a morphogen-induced mutual inhibition circuit from scratch that acts in isolation in *E. coli* and used it to investigate the conditions under which patterning occurs. We have also added morphogen production to the core circuit to create a reaction-diffusion patterning system that responds to a single gradient by producing two domains of gene expression with a self-organized boundary. The experimental control and precise measurement afforded by a synthetic biology framework allowed us to understand the behaviour of these patterning mechanisms at a quantitative level in the context of a mathematical model parameterized against data, and to uncover general design principles for engineering multicellular systems.

## Results

**Engineering mutual exclusivity**. To investigate whether a simple mutual inhibition network topology can interpret dynamic gradients, we built a synthetic Exclusive Receiver circuit (Fig. 1c), based on a previous Receiver circuit design (pR33S175[24]) that responds to two homoserine lactone (HSL) input signals, 3O-C6-HSL (C6) and 3O-C12-HSL (C12) with fluorescent protein outputs. We engineered mutual inhibition by introducing genes encoding TetR, expressed bicistronically with eYFP, and LacI, expressed bicistronically with eCFP. In addition, the C12-binding receiver protein LasR was expressed under the control of a LacI-repressible promoter, while the C6-binding receiver protein LuxR was expressed under the control of a TetR-repressible promoter. The Exclusive Receiver therefore consists of two signalling pathways that mutually repress each other, such that LasR, eYFP and TetR are expressed in the

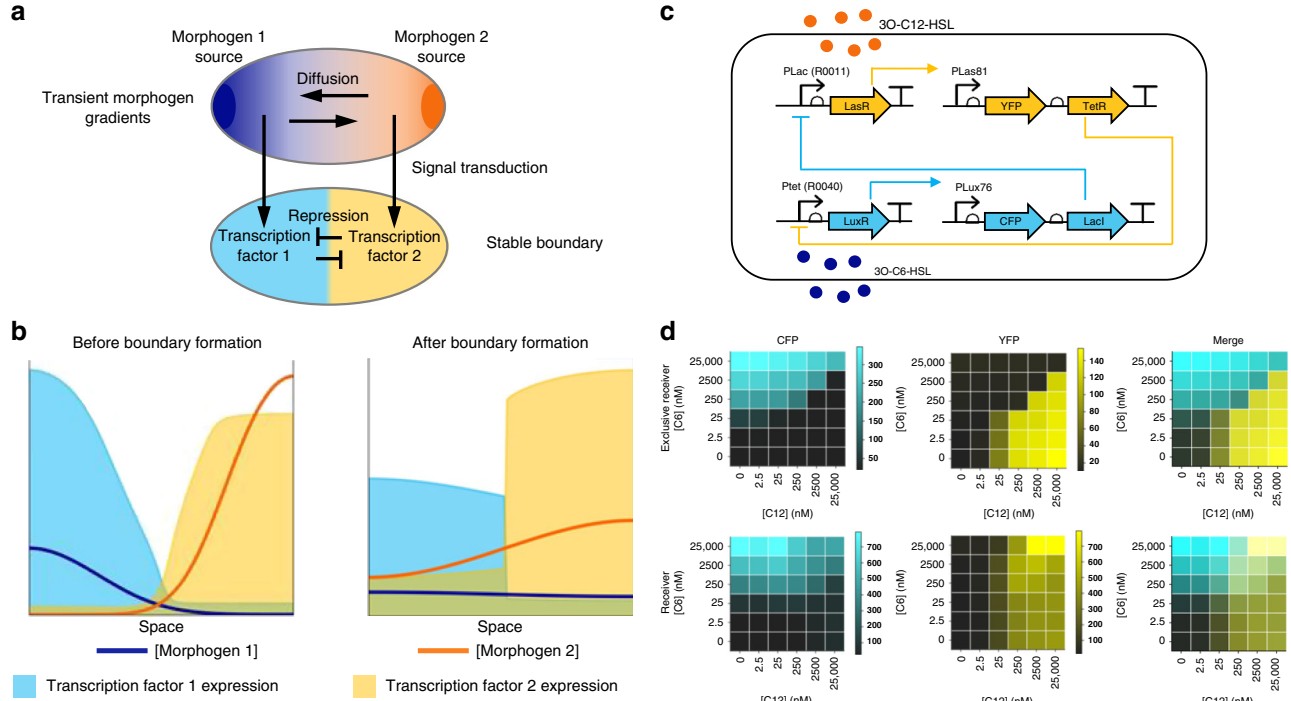

**Fig. 1 A synthetic gene circuit for morphogen interpretation. a** Schematic representation of a developing embryo. Mutual inhibition of transcription factors (cyan and yellow) downstream of antiparallel morphogen gradients (dark blue and orange) has been hypothesized to produce mutually exclusive domains of gene expression. **b** Morphogen gradients can be dynamic and transient, yet sharp, stable boundaries are observed between domains of gene expression. **c** A diagram of the Exclusive Receiver circuit. When 3O-C12-HSL (C12) levels are high, C12 binds to LasR, activating the expression of YFP and TetR, which represses the expression of LuxR, preventing expression of CFP and LacI. When 3O-C6-HSL (C6) levels are high, C6 binds to LuxR activating expression of CFP and LacI, which represses the expression of LasR, preventing expression of YFP and TetR. **d** Fluorescence output, measured in microplate fluorometer assays, of the Exclusive Receiver (top) and the Receiver (bottom) circuits represented as a ratio of CFP- (left) or YFP- (right) fluorescence to RFP fluorescence during exponential phase[38], cultured in the presence of the concentrations of C6 and C12 indicated. Data are representative of $n = 3$ biological replicate experiments conducted on different days. See Supplementary Figs. 11 and 12 for replicates. Source data are provided as a Source Data file.

presence of the signal C12, while LuxR, eCFP and LacI are expressed in the presence of the signal C6. Unlike the Receiver circuit, which responds to the presence of both signals by producing both fluorescent proteins, the Exclusive Receiver was designed to respond exclusively to the two signals, by producing either eCFP or eYFP. Mutually exclusive reception will only occur when the repressors (TetR and LacI) are expressed such that the level of repression produced by high concentrations of one HSL is sufficient to repress detectable quantities of the other and low concentrations of HSL produce little repression. We used this mutually exclusive response as a design goal for the Exclusive Receiver circuit. We constructed a range of designs and chose a variant that exhibited a strong difference between CFP- and YFP-expressing states while maintaining a roughly equal sensitivity to the two HSLs (Supplementary Fig. 1).

To characterise the response of the Exclusive Receiver circuit to varying concentrations of C6 and C12, we performed timecourse plate fluorometry assays and calculated the promoter activity from the CFP and YFP channels using a ratiometric method[25]. The circuit responded to high concentrations of C6 and C12 by producing CFP and YFP, respectively. As intended by the mutual inhibition design, mixtures of both signals resulted in only one fluorescent protein being produced, depending on the relative concentrations of the two signals (Fig. 1d; Supplementary Fig. 2). We confirmed these gene expression states at the single cell level using flow cytometry (Supplementary Fig. 3). In contrast, a Receiver circuit lacking mutual inhibition produced both fluorescent proteins simultaneously when both signals were present (Fig. 1d; Supplementary Figs. 2 and 3). Similar results were obtained when chemical inducers were used to suppress the repressors in the Exclusive Receiver circuit (Supplementary Fig. 4).

**Mutual inhibition results in bistability**. Central to the ability of mutual repression to produce a robust signal response is the

property of bistability, in which two stable steady states of gene expression are possible[20]. We first explored this property mathematically by developing a dynamic (ordinary differential equation) model of the Exclusive Receiver circuit. The model is based on one derived for the Receiver circuit[24], but incorporates the repressor proteins, TetR and LacI, and their regulation of LuxR/LasR expression (see Supplementary Methods for a complete derivation). We identified parameter values that enabled the model to reproduce timecourse fluorescence data using a previously established inference methodology in which a sequence of parameter inference tasks are applied to models and data for circuits of increasing complexity[26] (Supplementary Methods). This enabled us to simplify the identification of parameter values of the Exclusive Receiver model by reusing values of the subset of parameters that also appear in the Receiver model. We then applied numerical continuation methods to our data-constrained model to locate saddle-node bifurcations (see Supplementary Methods), and thus the concentrations of C12 and C6 for which bistability was predicted (Fig. 2a,b, red lines).

To test whether the Exclusive Receiver circuit exhibited hysteresis, a hallmark of bistability, at the concentrations predicted by the model, we first conditioned cells in either C6 or C12 and then exposed them to varied concentrations of both signals. At concentrations that produce bistability, we expected C6-conditioned and C12-conditioned cells to remain in the CFP-expressing and YFP-expressing states, respectively. In contrast, at concentrations that produce monostability, gene expression states would be determined solely by the final concentrations. We measured CFP and YFP expression by flow cytometry (Supplementary Fig. 5). The C6-conditioned cells expressed CFP at a wider range of concentrations, while C12-conditioned cells expressed YFP at a wider range of concentrations (Fig. 1b, Fig. 2a, b). We interpret this history-dependent difference in gene expression to be due to hysteresis. Thus, the region in concentration space in which we observe this difference is the

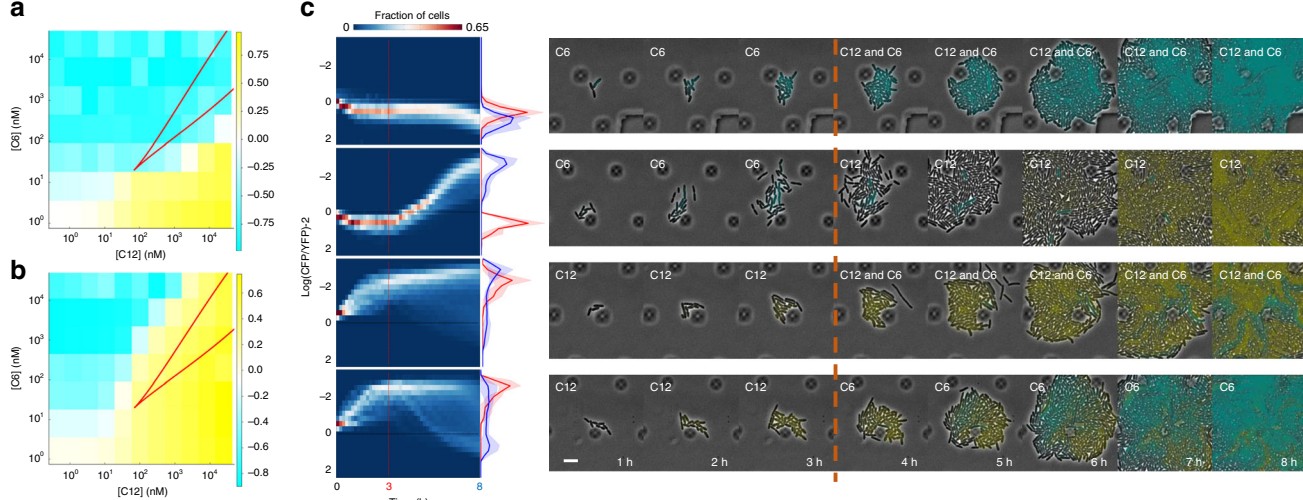

**Fig. 2 Mutual inhibition produces bistability.** Cells transformed with the Exclusive Receiver circuit were conditioned in either 500 nM C6 (**a**), or 500 nM C12 (**b**), and then exposed to the combinations of concentrations of C6 and C12 indicated. Cells were measured using flow cytometry and their normalized CFP minus YFP expressions were plotted. The region of bistability predicted by the parameterized model is the area within the red lines. See Supplementary Fig. 3 for gating strategy for all flow cytometry and Supplementary Fig. 13 for replicates. Source data are provided as a Source Data file. **c** Microfluidics cultures of cells transformed with Exclusive Receiver circuit in changing combinations of signals. Cells were grown for 3 h in the presence of either 37 nM C6 (rows 1 and 2) or 100 nM C12 (rows 3 and 4). Then media was changed to 100 nM C12 + 37 nM C6 (rows 1 and 3) or 100 nM C12 (row 2) or 37 nM C6 (row 4). Cells were imaged with a frame rate of (1 frame/10 min). Left panels are kymographs of the log-ratio of CFP expression per-cell to YFP expression per-cell, and fraction of cells as a heat map. Histograms represent the populations at 3 h (red) and 8 h (blue). Lines and shaded region represent the mean and standard deviation, respectively, over $n = 4$ biological replicates performed on 4 different days. Right panels are sample montages of cells switching state (rows 2 and 4) or exhibiting bistablity (rows 1 and 3); phase contrast and fluorescence channel ranges chosen for display. Scalebar = 6 μm.

region of bistability. This region was slightly larger than that predicted by the model but was qualitatively similar, which suggested that our model captured the essential features of our circuit.

To determine whether individual cells were switching or retaining their gene expression states in response to signal combinations, as suggested by our flow cytometry data, we used microfluidic devices to observe the dynamics of single cells over time. We cultured cells with 37 nM C6 or 100 nM C12, concentrations expected to produce bistability in combination, and then switched to the opposite signal. We found that cells with high fluorescence expression could, indeed, switch to the alternative state (Fig. 2c rows 2 and 4, Supplementary Fig. 6), even when we used the much higher concentration of 1 μM C6 (both to initially condition cells and to switch their state Supplementary Figs. 7 and 8). When cells conditioned with either C6 or C12 were exposed to both signals, the cells exhibited bistable behaviour, mostly maintaining their initial fluorescence states (Fig. 2c rows 1 and 3, Supplementary Figs. 6–8) although a comparatively small population of C12-conditioned cells do begin expressing CFP in response to mixed signals (Fig. 2c, row 3) indicating some heterogeneity in the population with respect to bistability. We hypothesize that cell to cell differences in gene expression result in differences in the HSL concentration regimes in which the cells exhibit bistability resulting in heterogeneity in switching behaviour. Together, these data indicate that, when signal concentrations gave rise to bistability, the final gene expression state was determined by the history of exposure to signals.

**Hysteresis produces stable boundaries**. To test how the Exclusive Receiver circuit interpreted dynamic morphogen gradients, we grew cells on filter paper printed with hydrophobic ink, such that growing colonies remained within the square in which they were inoculated but signals could diffuse through the agar beneath the filter[24,27]. We performed these experiments in agar containing 10 μM IPTG (1% of the standard induction concentration of 1 mM) to create a regime in which both CFP-dominating and YFP-dominating regions were observed. The addition of IPTG was required to compensate for differences in culture conditions between solid and liquid cultures (see Supplementary Fig. 34 for experiments without IPTG). While the precise mechanism is unknown, differences in culture conditions when switching to solid culture appeared to shift the bistability region such that even very low concentrations of C6 enabled bistability, preventing the appearance of a YFP-dominant regime. This was redressed through addition of IPTG, which partially derepresses LacI, shifting the region to coincide with the region in liquid culture. We cast either C6 or C12 into the agar at each end of the filter paper at varying concentrations and performed timelapse imaging of the fluorescence output of the circuit in response to the dynamic gradients produced by diffusion. When C6 and C12 were provided at concentrations that, if allowed to diffuse to homogeneity would result in 200 nM C6 and 2000 nM C12, (i.e., a 200 nM and 2000 nM spatial average, respectively), a sharp boundary was produced between mutually exclusive domains of CFP and YFP (Fig. 3a). Plotting the point in space at which CFP and YFP expression were equal (see Supplementary Methods) against time revealed that the boundary between domains did not move over time (Fig. 3b). In contrast, when spatial average concentrations of 20 nM C6 and 2000 nM C12 were used, there was an initial production of CFP near the source but it was quickly overwhelmed by YFP production and the point of equal expression moved toward the C6 source (Fig. 3a). These images measured bulk (rather than per-cell) fluorescent protein

expression so highly stable fluorescent proteins would remain detectable even after they stopped being expressed. It was therefore unsurprising that CFP remained detectable in cells close to the C6 source due to protein perdurance. Importantly, however, the CFP intensity did not increase, whereas the level of YFP did increase, indicating that the circuit had switched from CFP expression to YFP expression in these cells. At varying combinations of signal concentrations we classified boundaries (see Supplementary Methods) as static (S), moving (M), or not present (N) (Fig. 3c, Supplementary Fig. 9, Supplementary Movie 1) and observed that static boundaries occurred at signal concentrations similar to those that produced hysteresis (Fig. 2a and b).

To understand this behaviour we performed simulations and analysis of our mathematical model. We plotted the concentration of C6 and C12 experienced by physically separated cells and visualized the changes in concentration that they experienced due to diffusion (Fig. 3d, Supplementary Movie 2). We observed that cells at different points in space all converge to the spatial average concentration, but they do so by taking different paths, resulting in differences in CFP and YFP expression. Points closest to the C6 source begin in the monostable CFP region, whereas those closest to the C12 source begin in the monostable YFP region. This means that the cells are traversing the bistable region along different paths and will therefore exhibit hysteresis as they converge to the spatial average. If the spatial average concentrations lie within the bistable region, which is defined in concentration space, all cells will eventually experience concentrations within this region as the morphogens diffuse. Thus, over time, the region of cells in physical space that exhibit bistability expands to encompass the entire domain (Fig. 3e, S; Supplementary Movie 3). The result is that a cell's state will be determined by its history and cells with different histories that originated on different sides of the boundary will end up in different stable states. The cells close to the C6 source will behave like cells conditioned in C6 and express CFP while the cells close to the C12 source will behave like cells conditioned in C12 and express YFP (as in Fig. 2) and will maintain their states even after the two morphogens mix via diffusion. The result is the formation of two mutually exclusive domains of gene expression with a sharp boundary that is stable and stationary, even though the morphogen gradients that produced those domains were only present transiently. If, on the other hand, the spatial average concentrations lie outside the bistable region, a transient boundary will form and cells will switch fates as they leave the bistable region, taking on the fate determined by the morphogen of greater concentration (Fig. 3e, M; Supplementary Movies 4 and 5).

**A secondary gradient creates self-organised domains**. Given the ability of the mutual inhibition topology to produce stable domains of gene expression in response to antiparallel morphogen gradients, we hypothesized that it could function similarly in response to a single morphogen gradient with the addition of a secondary gradient produced by the cells themselves, which functions as a lateral inhibitor[28]. This circuit mimics the sequential induction of organizing centres found in *C. elegans* vulval development[29], the rhombomeres of the vertebrate hindbrain[30,31], and the *Drosophila* wing disc[32].

To explore this mechanism we added previously characterized Relay circuits[24], which produce one signal in response to the other, to the Exclusive Receiver circuit. This created an Exclusive Relay circuit that both produces and interprets morphogen gradients (Fig. 4a and Supplementary Methods). We created a transient gradient of C6 by replacing a cylinder of agar in the centre of a plate with agar containing 40 μM C6, and plated cells

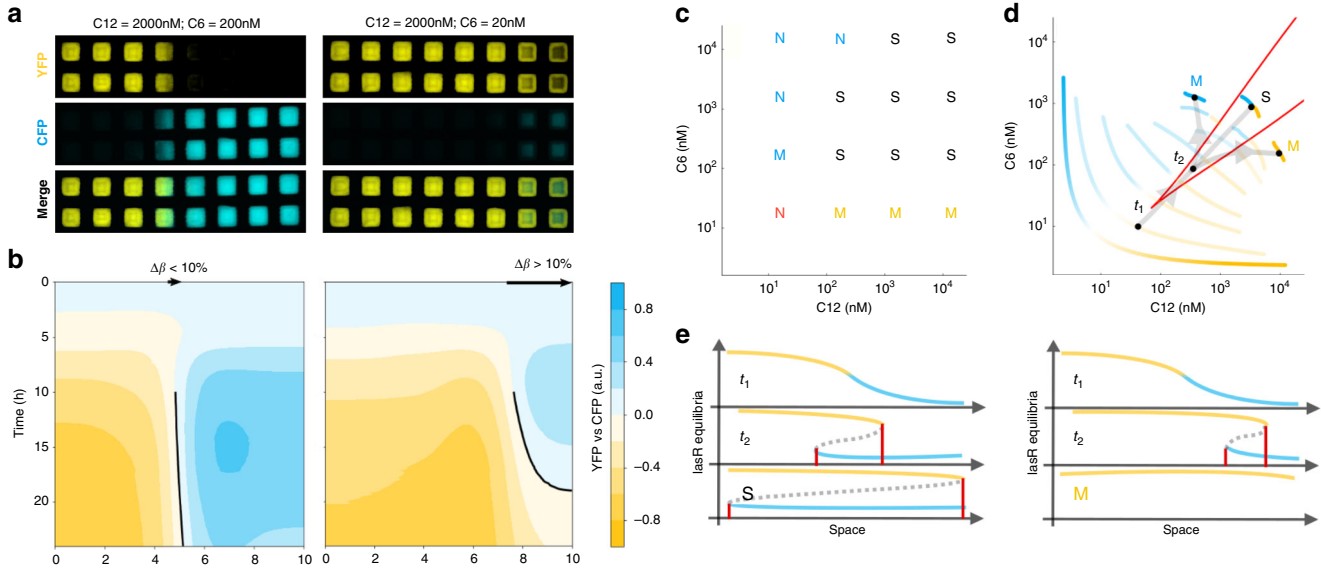

**Fig. 3 Formation of stable boundaries. a** Endpoint fluorescence microscopy of Exclusive Receiver cells grown in transient gradients of signals (C12 diffusing from the left, C6 diffusing from the right) at the spatial average concentrations indicated and in the context of 10 μM IPTG throughout. Representative examples (n = 3 biological replicates performed on 3 different days) of a static boundary (left) and a moving boundary (right) **b**, Corresponding kymographs of CFP and YFP fluorescence (intensity) over time (y-axes, hours) at different spatial positions (x-axes, mm). If the location of the boundary (location of equal normalized CFP and YFP fluorescences, black lines) at the end of the timelapse minus its location when it became detectable ($\Delta\beta$, arrows) was less than 10% of the domain size we considered the boundary stable. **c** Boundaries were evaluated as above at the signal concentrations indicated by letters. S indicates equilibrium concentrations at which static boundaries were observed. M indicates a moving boundary. "N" indicates no boundary. The colour of the letter indicates which FP was dominant and red indicates neither FP dominant. See supplementary Figs. 31–33 for replicates. **d** Schematic representation of the concentrations of C6 and C12 experienced by cells at different points in physical space (cyan and yellow curves) as gradients diffuse to homogeneity. Paler curves represent different timepoints. If the spatial average concentrations lie within the region of bistability, the boundary will be static (S), otherwise the boundary will move (M) and will eventually be abolished as cells adopt either CFP or YFP expression. $t_1$ and $t_2$ indicate timepoints considered in **e**. **e** Corresponding schematic representing LasR expression, coloured according to resultant fluorescent protein expression. Dashed line indicates the location of an unstable local equilibrium. Red lines indicate the spatial location in which cells are exhibiting bistability. In the case of a stationary boundary (S), the region of space containing cells exhibiting bistability expands to encompass all cells and their gene expression state is determined by their history. In the case of a moving boundary (M), the region exhibiting bistability moves rightward and disappears and the domain becomes dominated by a single monostable state.

transformed with the Exclusive Relay circuit onto gridded filters. The primary C6 gradient resulted in cells in the centre expressing CFP, LacI, and LasI (Fig. 4b and c). These cells produced C12 but were unable to sense it because they did not express LasR due to its repression by LacI. The gradients of C6 and C12 overlapped but the C12 gradient could extend further due to C12 being actively produced by a large region of cells (Fig. 4d). At a certain distance from the source of C6, the ratio of C12 to C6 favoured the C12 state such that the bistable switch 'flipped' and cells expressed YFP and TetR, repressing LuxR and the ability to sense C6. The result was two domains of mutually exclusive gene expression from a single primary morphogen gradient. As in the case of antiparallel gradients, the stability of the boundary between these domains of gene expression can be understood using the same framework: The spatial average concentration of the primary morphogen remained constant as the total density that was added at the beginning of the experiment was unchanging. The secondary morphogen, in contrast, was being produced by cells so the total density increased over time. The spatial average of both morphogens therefore moved along the axis of the secondary morphogen as cells that were sensing the primary morphogen produced the secondary (Fig. 4e, red arrows). The result is that a metastable boundary is produced that is present as long as the system remains within the region of bistability but will eventually be abolished as the secondary morphogen accumulates. Transforming the Exclusive Receiver

with a P81-LuxI Relay and creating a C12 primary gradient resulted in equivalent patterning with the physical location of the states reversed (Supplementary Fig. 10).

## Discussion

By building a synthetic gene circuit composed of mutual inhibition downstream of diffusible morphogens, we have shown that this network topology in isolation is sufficient to recapitulate the behaviour, seen in multicellular developmental systems, of mutually exclusive domains of gene expression separated by a boundary that is sharp and stable despite transient and dynamic morphogen gradients. This topology also proves to be robust to differences in morphogen concentration, as variations in concentration spanning orders of magnitude result in only small changes in boundary location (Supplementary Fig. 9, Supplementary Movie 1). In addition, we have built a patterning circuit that creates a self-organised boundary between two gene expression domains in response to a primary morphogen by creating a secondary morphogen that functions as a lateral inhibitor. This circuit shares features of both a positional information model[1], in that it interprets a preexisting morphogen gradient to produce domains of gene expression, and a reaction-diffusion model[33,34], in that morphogen production is coincident with interpretation. These two models have previously been presented in opposition to each other, but it is likely that both mechanisms are at work in development[35]. Our Exclusive Relay

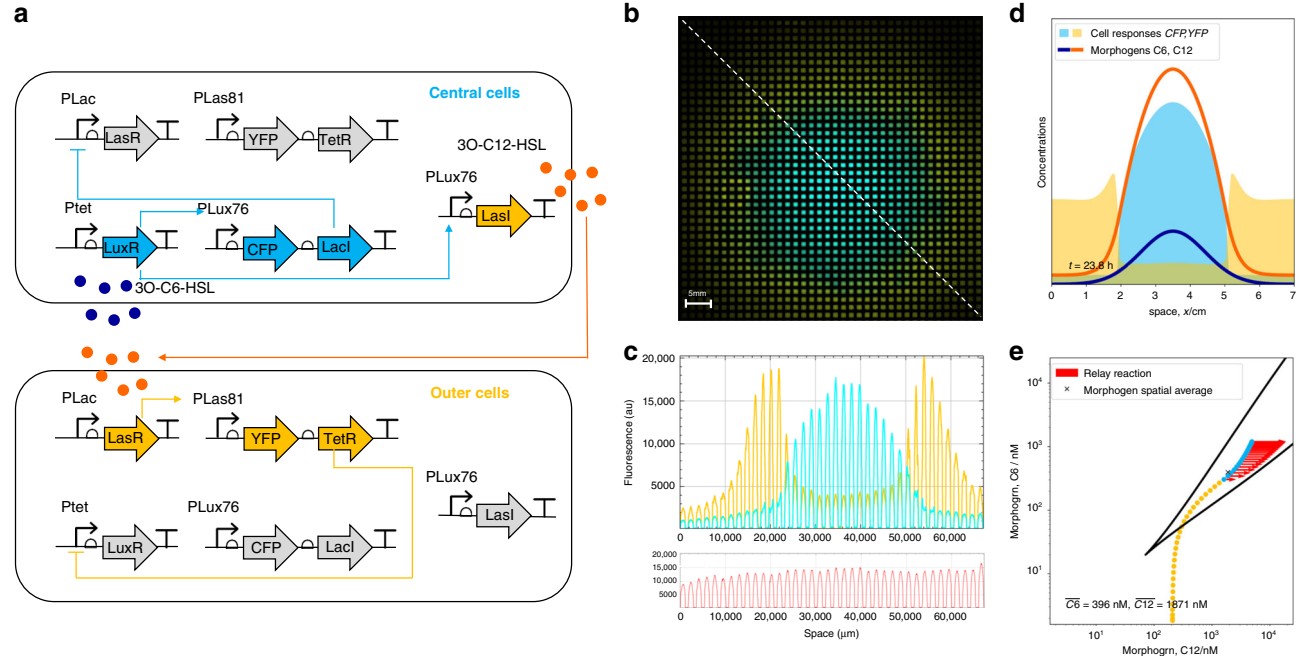

**Fig. 4 Addition of a relay circuit creates self-organised domains of gene expression. a** Circuit diagram of Exclusive Receiver cells co-transformed with a Relay circuit (P76-LasI) that responds to C6 by producing C12. **b** Isogenic cells transformed with the circuit shown in **a** and grown for 24 h in the presence of a gradient of C6 diffusing from the centre. Cells that experience high levels of C6 (central cells) will express CFP, LacI, and LasI, causing them to produce C12 but be unable to sense it. Neighbouring cells (outer cells) that do not experience C6 will sense C12 and express YFP and TetR, resulting in mutually exclusive domains of gene expression. Cells also constitutively express mRFP1 via a genomic transgene. Image is representative of 3 biological replicates performed on 3 different days. **c** Quantitation of fluorescence along the dotted line in **b**. Cyan, yellow, and red indicate CFP, YFP, and RFP expression, respectively. **d** Final timepoint of simulation shows a secondary gradient of C12 (orange) produced in response to the primary C6 gradient (dark blue). Cyan and yellow indicate simulated CFP and YFP expression, respectively. **e** Final time point of simulation in C6-C12 space labelling points in physical space by their CFP and YFP expression (cyan and yellow points), and showing the production of C12 as vectors (red arrows) that move the spatial average (x) toward increasing C12 (see supplementary video 6).

circuit forms patterns by using two in-phase morphogen gradients to produce a primary response whose spread is limited by the gradient of the secondary morphogen, due to its larger magnitude and diffusive radius. The result is concentration-dependent domains of gene expression produced in response to a morphogen gradient, just as in the French flag model. However, due to the hysteresis of the bistable core of the circuit, these domains of gene expression are metastable even though the primary morphogen gradient diffuses to homogeneity. It is worth noting that the boundary between the eYFP-expressing domain and the domain that expresses neither fluorescent protein is determined simply by the threshold of response to C12 and is therefore not a stable boundary. This boundary could be stabilised by recapitulating the mechanism we have described via the addition of a third morphogen (and mutual inhibition with C12) either diffusing from the opposite direction as the primary gradient (as in Fig. 3) or as a second relay mechanism (as in Fig. 4). The fact that genetic circuits optimized in different contexts can be directly composed to produce more complex patterns suggests that the synthesis of reaction-diffusion and positional information mechanisms may be readily obtainable through evolution, and therefore common in development. This also provides a blueprint for designing synthetic gene circuits that produce spatiotemporal patterns in cell populations, which could lay the groundwork for rationally designing self-organizing, self-repairing materials and tissues.

## Methods

**Plasmid construction.** The exclusive receiver circuit and variants described in Supplementary Fig. 1 were cloned using Gibson Assembly[36] using pR33S175[24] as a starting point. Primers used for Gibson assembly can be found in Supplementary Table 1.

**Plate fluorometer assays.** The exclusive reporter construct was transformed into EC10G *E. coli* cells with a chromosomally integrated mRFP1 construct[24]. Overnight cultures were grown from glycerol stocks in M9 media supplemented with 0.4% glucose, 0.2% casamino acids, and 50 μg/ml kanamycin (supplemented M9) then diluted back 1:100, allowed to grow to an OD of 0.3 then diluted 1:1000. Cultures were aliquoted into black-walled, clear-bottom 96-well plates (Greiner μClear) in a volume of 200 μl per well and measurements taken every 10 min for ~1000 min in a BMG FLUOstar Omega plate fluorometer using BMG FluoStar Omega Reader Control Software 5.10R2. 3-oxohexanoyl-homoserine lactone, (C6, Cayman Chemicals) and 3-oxododecanoyl-homoserine lactone C12, Cayman Chemicals were dissolved to a concentration of 200 mM in DMSO then C6 was diluted in supplemented M9 to the concentrations described, while C12, due to its limited solubility in aqueous media, was first diluted 1:50 in ethanol then diluted in supplemented M9 medium to the concentrations described.

**Flow-cytometric analysis of hysteresis.** EC10G cells transformed with the exclusive receiver construct were grown overnight from glycerol stocks as described for plate fluorometer assays. Overnight culture was diluted 1:100 and incubated until OD 0.2. Cells were then resuspended 1:100 in supplemented M9 supplemented with either C6 or C12 at 500 nM each and conditioned for 2 h. Following conditioning, cells were washed three times in supplemented M9 by centrifugation at 3200 × *g* for 4 min. and seeded at 1:1000 into wells of a 96-well plate containing combinations of varying concentrations of C6 and C12 as indicated in Supplementary Fig. 5. The plate was incubated for 5h with continuous OD monitoring. OD measurements at 5 h were consistently within the exponential growth range (0.3–0.8). Following 5 h incubation, cells were diluted 1:6 in PBS and analysed by flow cytometry on a BD FACSCelesta (BD Biosciences, San Jose, CA, USA) equipped with HTS and a standard optical setup. Data was collected using FACSDiva 8.01. CFP was excited with violet laser 405 nm and detected with 525/50 BP filter - 505 LP mirror combination. YFP was excited with blue laser 488 nm and detected with 530/30 BP filter - 505 LP mirror combination. RFP was excited with yellow-green laser 561 nm and detected with 610/20 BP filter - 600 LP mirror combination. Instrument quality control was performed prior to each experiment

using BD CS&T beads. Fluorescence compensation parameters were determined using induced and untreated exclusive reporter cells and 30,000 events were counted within RFP gate for each sample. Data analysis was performed with FCS Express v.7 software (DeNovo Software, Glendale, CA, USA). The gating strategy for all flow cytometry was the same and is shown in Supplementary Figs. 3c and 5c.

**Microfluidics.** Glycerol stocks of EC10G transformed with the Exclusive Receiver were streaked onto LB agar plates. Colonies were picked into M9 and grown at 37 °C overnight, then diluted 1:1000 into M9 and grown for 4 h 45 min at 37 °C into exponential phase. The culture was typically diluted 1:100 in M9 before being loaded into the CellASIC ONIX B04A-03 microfluidic device using the manufacturer's protocol (EMD Millipore Corporation). Cells were supplied with media using a pressure of 5 psi in the device. The entire device, along with most of the microscope, was incubated at 37 °C during movie acquisition. Cell segmentation was done on the RFP channel using the published Schnitzcells software (release 1.1, 2005)[37]. The mean YFP and CFP fluorescence normalized to cell area was then calculated by averaging the corresponding pixels in the respective channels. Movies with no cells or non-growing cells were excluded by keeping only those movies with greater than 20 and 50 cells at 3 and 6 h, respectively. Non-cell segmentation artefacts were excluded by area (<200 pixels) and Euler number (<1) computed with the regionprops function of MATLAB 2014a.

**Microfluidics microscopy.** Microfluidics devices were imaged using a widefield microscope with epifluorescence and phase contrast imaging modes (Nikon Ti-eclipse, Nikon, UK) equipped with the Nikon Perfect Focus (PFS) Unit. Illumination for the epifluorescence was provided by a white light LED source (SOLA SE Light Engine or Spectra X Light Engine, Lumencor, USA), transmitted by a liquid light guide (Lumencor, USA), through a fluorescence filter cube (YFP Channel: 49003: ET/Sputtered series ET-EYFP, exciter: ET500/20x, dichroic: T515LP, emitter: ET535/30m; CFP Channel: 49001: ET/Sputtered series ET-CFP, exciter: ET436/20x, dichroic: T455LP, emitter: ET480/40m;  RFP Channel: 41027-Calcium Crimson, excitation: HQ580/20x, dichroic: Q595LP, emitter: HQ630 /60m, Chroma, USA), and a CFI Plan Apochromat 100x oil immersion objective (NA 1.45, Nikon). Phase contrast illumination was provided by a 100 W lamp via a condenser unit (Nikon). Images were acquired on a CoolSNAP HQ2 camera (Photometrics, USA). The sample was held in motorized stages (Nikon). The sample was incubated along with much of the microscope body using a temperature controlled, heated chamber (Solent Scientific, UK). The microscope was controlled with MetaMorph software (version 7.8.10.0, Molecular Devices, USA). Fluorescent beads (TetraSpeck microspheres, 0.5 um, Molecular Probes, USA) were imaged as a calibration standard.

**Solid culture assays.** Exponential phase cultures were grown to an OD of 0.3 and plated onto Iso-Grid membranes (Neogen) on supplemented M9 with 1.5% agar at a volume of 0.5 μl per square. Gradients were created by cutting holes in supplemented M9-agar (1.5%) plates (cast in OmniTray [Nunc]) containing 10 μM IPTG. Holes were cut on both ends of a domain to be inoculated at a size of 25% of the domain, each. Holes were then filled with liquid M9-agar to which either 3O-C6- or 3O-C12-HSL had been added at 4X concentration. After hardening, excess agar was cut away leaving each domain isolated. For relay circuit assays, circular holes were punched in the centre of plates using the back of a pipette tip and the holes were filled with ~200 μl of liquid agar containing 40 μM of the appropriate HSL. Plates were sealed with parafilm and imaged using a motorized Leica M205 FA fluorescence stereo microscope controlled using Leica LAS X software. Plates were incubated at 37 °C using a DigiTherm microscope temperature control air bath (Tritech Research). Illumination was an LED white light source (Lumencor) with excitation filters of 426–446 nm, 490–510 nm, and 555–589 nm, and emission filters of 460–500 nm, 520–550 nm, and 608–682 nm. Tiled images were taken every 10 min and were stitched using Leica LAS X software.

**Reporting summary.** Further information on research design is available in the Nature Research Reporting Summary linked to this article.

## Data availability

Source data are provided with this paper. Plate fluorometer and flow cytometry datasets can be found in supplementary file sourcedata.zip. Raw microscopy images (Figures 2c, 3a, 4b) are available on request to the authors. The exclusiver receiver plasmid is available from Addgene (Addgene ID 160376). All other relevant data are available from the authors upon reasonable request. Source data are provided with this paper.

## Code availability

Code is available at GitHub repository https://github.com/gszep/double-exclusive-reporter. Source data are provided with this paper.

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

## Acknowledgements
We would like to thank Jeremy Green for helpful comments on the manuscript.

## Author contributions
P.K.G., N.D., J.H., and A.P. conceived and designed the study. P.K.G. designed and built the genetic circuits. P.K.G., O.P., and V.C. performed the experiments. G.S., J.H., and N.D. conceived and implemented theory and modelling and wrote the supplementary information. All authors analysed and interpreted the data. P.K.G. and A.P. wrote the main text. All authors provided input into the manuscript.

## Competing interests
The authors declare no competing interests.
