## [Peer Review File · Nature Communications]

Reviewers' Comments:

Reviewer #1:

Remarks to the Author:

The authors have done a fine job in responding to my comments on the technical aspects of the work. As stated in my initial review, I do believe the work is technically sound and presents comprehensive data to show how a bistable switch can allow generation of sharp boundaries in response to a spatial gradient of chemicals.

On the conceptual side, the authors argue for the significance of stable boundary which I do notice. However, that is expected from the capability of generating hysteresis by a bistable switch. Ultimately, it boils down to the ability of a bistable switch to lock into a stable state after experiencing a transient stimulation, in a history dependent fashion.

Therefore, I think the conceptual significance of the work is overstated.

Please see below the reviewer's comments with our responses in red:

The authors have done a fine job in responding to my comments on the technical aspects of the work. As stated in my initial review, I do believe the work is technically sound and presents comprehensive data to show how a bistable switch can allow generation of sharp boundaries in response to a spatial gradient of chemicals.

We thank the reviewer for their compliments on the technical aspects of the work and on how our data supports the conclusions of the paper.

On the conceptual side, the authors argue for the significance of stable boundary which I do notice. However, that is expected from the capability of generating hysteresis by a bistable switch. Ultimately, it boils down to the ability of a bistable switch to lock into a stable state after experiencing a transient stimulation, in a history dependent fashion.

We agree with the reviewer's description of the mechanism by which bistability produces stable boundaries and we think it is important to demonstrate experimentally how such a mechanism can play out in a real system.

Therefore, I think the conceptual significance of the work is overstated.

We have revised the abstract to clarify the conceptual significance of the work as requested by the reviewer, highlighting the experimental validation of the theoretical mechanism.